# Chronic Vulvar Pain and Health-Related Quality of Life in Women with Vulvodynia

**DOI:** 10.3390/life13020328

**Published:** 2023-01-24

**Authors:** Gabriela Patla, Agnieszka I. Mazur-Bialy, Magdalena Humaj-Grysztar, Joanna Bonior

**Affiliations:** 1Department of Medical Physiology, Chair of Biomedical Sciences, Institute of Physiotherapy, Faculty of Health Sciences, Jagiellonian University Medical College, 12 Michałowskiego Street, 31-126 Krakow, Poland; 2Department of Biomechanics and Kinesiology, Chair of Biomedical Sciences, Institute of Physiotherapy, Faculty of Health Sciences, Jagiellonian University Medical College, 8 Skawińska Street, 31-066 Krakow, Poland; 3Institute of Nursing and Midwifery, Faculty of Health Sciences, Jagiellonian University Medical College, 58 Zamoyskiego Street, 31-501 Krakow, Poland

**Keywords:** vulvodynia, chronic vulvar pain, VAS, quality of life, WHOQOL-BREF

## Abstract

**Simple Summary:**

Vulvodynia is defined as chronic vulvar pain in the perineal area. It affects women around the world and is a significant health problem. The authors of this study examined the intensity of vulvar pain and the impact of vulvodynia on quality of life in a group of Polish women. It was observed that vulvodynia mainly occurs in young people with a high level of education, and predominantly in unmarried women, divorcees, and widows. In addition, the disease reduces quality of life, which is mainly due to difficulties in performing activities of daily living and a decrease in sexual satisfaction. The more pain a woman feels, the worse her quality of life. It was concluded that vulvodynia is a major problem among the Polish female population. Thus, research in this regard should be continued by scientists and health care professionals.

**Abstract:**

The aim of this study was to investigate the severity of chronic vulvar pain in women with vulvodynia and its impact on their health-related quality of life (QL). The study group consisted of 76 women aged 19 to 58. The study was carried out using the diagnostic survey method, i.e., (1) the questionnaire technique, comprising (A) the author’s questionnaire (76 questions) and (B) the WHOQOL-BREF questionnaire, and (2) the VAS. When analyzing the severity of vulvar pain on the VAS, the highest proportion of women rated it at level 6 (23.68%). This was significantly determined by certain personal characteristics (age < 25 years old) and sociodemographic characteristics (marital status: unmarried women, divorcees, widows; high school education), each at *p* < 0.05. Vulvodynia causes a significant deterioration (64.47%) in QL, which is mainly caused by a reduction in the ability to perform activities of daily living (27.63%) and a decrease in sexual satisfaction (27.63%). The level of stress significantly exacerbates pain (*p* < 0.05). The severity correlates significantly (*p* < 0.05) and negatively (r < 0) with QL perception, which was rated worst in the physical domain. The use of treatment resulted in a significant improvement in the physical and psychological domains (*p* < 0.05), and the latter was particularly influenced by physiotherapy (*p* < 0.05).

## 1. Introduction

Vulvodynia (lat. vulva “vulva” and stgr. ὀδύνη odynē “pain”) is the term used to describe vulvar pain lasting at least 3 months without a clear, organic, identifiable cause, which may have potential provoking factors [1,2]. The disease significantly affects patients’ and their families’ quality of life and biopsychosocial status.

The prevalence of the disease in populations around the world has not yet been well studied, and when making statistical comparisons, large discrepancies are apparent. However, on the basis of the most recent data, it is estimated that, in general, it affects 8 to 12% of women [3]. This, for example, ranges from 5 to 16%, or up to 6 million women, in the United States of America. Unfortunately, in Poland, as a result of a lack of sufficient studies, exact data on this subject are not available [4,5,6,7,8]. However, as a result of the complexity of vulvodynia, it is presumed to be a condition that is “underdiagnosed”, and the percentage of sufferers worldwide is supposed to be higher [9]. Importantly, the condition occurs in women of all ethnic groups and ages [10].

The etiopathogenesis of the disease is not completely understood, and diagnosis is based on the exclusion of other organic causes, which were highlighted in the 2015 consensus (Table 1) [2].

Myofascial dysfunctions involving the pelvic diaphragm are found in women with vulvodynia. These are usually in the form of increased tension, which is termed a hypertonic/overactive/nonrelaxing state and can be a primary or a secondary (trauma, infection, endometriosis) cause of pain [7,8,9]. The disorder mainly affects the largest muscle of the pelvic floor, the levator ani, but also spreads to the anterior and/or posterior muscular segments, and can cause dysfunction of the neighboring organs and systems, consequently leading to dysfunction of the bladder, rectum, and even intestines. Accordingly, women with vulvodynia are found to have characteristic comorbidities involving the urinary bladder (overactive or painful organ, or the surrounding area; recurrent or interstitial inflammation), micturition disorders (urgent incontinence; difficulty initiating micturition or thin micturition stream), rectum (hemorrhoids; anal fissure; pruritus ani; pain and problems with defecation), or irritable bowel syndrome. However, it has also been shown that there are clinical cases in which myofascial dysfunctions not only affect the pelvic floor muscles, but also other areas of the body. Women experience pain in the neck and the lumbosacral region, and headaches (including migraines), tension in the temporomandibular joint area, and fibromyalgia are also sometimes observed [11].

An important group of risk factors, which can have a significant role in the development of pelvic floor dysfunction, are those relating to the psychosocial sphere. Anxiety, depression, catastrophic thinking, and post-traumatic stress often lead to vulvar pain and sexual dysfunction. The reduced effectiveness of analgesic treatment is found in this group of individuals. In addition, women with anxiety disorders have been shown to have as much as a fourfold increased risk of vulvodynia [12,13,14].

According to the World Health Organization’s (WHO) guidelines, chronic pain is a disease entity, and the patient is entitled to receive full therapeutic care [15].

On the basis of the localization of pain symptoms, vulvodynia is divided into:A.Generalized (which involves the entire vulva, and may spread to other areas of the body, such as the abdomen, thighs, or buttocks);B.Localized (which involves only part of the vulva)—subtypes:
Clitorodynia (clitoral pain);Westibulodynia (pain in the vaginal vestibule);Hemivulvodynia (pain of the middle of the vulva).
C.Mixed [16].

The duration and nature of vulvar pain varies among women and can be intermittent, unchanging, continuous, immediate, or delayed, and sensations are completely subjective, being described as itching, burning, stinging, irritation, dryness, abrasion, and hypersensitivity of the vulvar area [5,10]. The pain can occur without or under the influence of external factors, and forms are distinguished as follows:A.Spontaneous;B.Provoked;C.Mixed [2].

Dyspareunia, or pain during sexual intercourse, is classified as a provoked form of the disease. Other triggers or situations that cause the provoked form of the disease can be wearing clothing that is too tight, an attempt to apply a tampon, a gynecological examination, and even prolonged sitting.

The etiopathogenesis of the disease makes the therapeutic process complicated and time-consuming, hence it is based on several concepts that complement one another. The woman should remain after multidisciplinary care. Studies show that such a therapeutic path yields the best results. Treatment methods include pharmacotherapy, surgical treatment, psychotherapy, sex therapy, and a set of physiotherapeutic methods that now seem to be expanding rapidly with promising results, and are recommended by the American College of Obstetricians and Gynecologists [8,17,18,19,20,21,22,23].

Techniques that have been confirmed in studies include manual techniques, e.g., relaxation of tense pelvic floor muscles; mobilization of the fascia and joints; work on local trigger points in the pelvic floor; pelvic floor muscle exercises; stretching exercises, and exercises with biofeedback control; relaxation and breathing exercises; and the use of TENS treatment [8,10,18,24,25,26].

Vulvodynia as a chronic disease poses a serious challenge to health care in both diagnostic and therapeutic terms, and in psychological, social, and economic terms. Hence, it is very important to measure and monitor quality of life (QL), which is particularly applicable to medicine and health care more broadly, and pain severity, so that the therapeutic team can effectively and comprehensively manage these patients. In the course of vulvodynia, there is a chronification of the disease process and the need to deal with its difficult consequences. All this prompts the need for multidirectional research that will contribute to improving the QL of women struggling with chronic vulvar pain, which often lasts for several years and impinges on the overall health and social relations, including the all-important intimate contact with the partner, the emotional state, and, often, the economic status of sufferers [27]. A QL assessment should be considered on par with medical indicators of health.

Quality of life (according to the WHO) is described as an individual’s subjective assessment of his or her life situation in relation to the culture in which he or she lives and in relation to their goals, expectations, interests, and professed life values. This general definition takes into consideration all aspects of a person’s life multidimensionally, and in order to systematize the issue, the concept of health-related quality of life (HRQOL) was introduced into medicine. This term is closely related to the WHO’s definition of health, which it describes as “a state of complete physical, mental and social well-being and not merely the absence of disease” [28].

There are a number of standardized tools for assessing QL, including HRQOL. The most common scales in the literature are the Skindex-29, the EuroQol-5-dimension questionnaire (EQ-5D), and the Dermatology Life Quality Index (DLQI). Skindex-29 is a skin-specific questionnaire that has 29 items divided into domains (physical symptoms, functioning, and emotions). Each item is scored on a 5-point Likert scale. All responses are transferred to a linear scale. The overall score is the average of the responses [29]. The EQ-5D questionnaire is a tool that describes five dimensions: mobility, self-care, usual activities, pain/discomfort, and anxiety/depression. Each dimension has five levels: no problems, minor problems, moderate problems, severe problems, and extreme problems. The second part, the so-called EQ VAS, is a vertical visual analog scale that assesses a patient’s current health status. The DLQI is a life index dependent on skin complaints. Its purpose is to measure the extent to which skin complaints have affected the patient’s life in the past week. It consists of 10 questions that relate to ailments, activities of daily living, interpersonal relationships, ability to work or study, and sex life. The patient answers on a scale from 0, i.e., not at all, to 3, i.e., very much [30]. However, these are not the only questionnaires that can be used. Another tool that accurately examines QL in vulvodynia is the World Health Organization Quality of Life (WHOQOL-BREF) standardized questionnaire, which assesses quality of life in six dimensions: perception of quality of life and health and quality of life in the physical, mental, social, and environmental domains. It is worth noting that, in other studies in the physiotherapy field, the visual analogue scale (VAS) is used to assess the severity of pain determining HRQoL, which can be especially helpful when monitoring treatment effects. Examples include foot dysfunction [31], idiopathic scoliosis [32], and total hip or knee replacement [33].

Taking into account the prevalence of vulvodynia and the underestimation resulting from diagnostic difficulties, it is important to conduct research to better understand this disease. Various studies indicate that vulvodynia significantly affects all aspects of a woman’s life, significantly reducing quality [8,34,35]; however, to the best of our knowledge, the Polish population has yet to be assessed. The aim of this study was to investigate the severity of chronic vulvar pain in women with vulvodynia and its impact on their HRQoL. We hypothesized that the HRQoL of women with vulvodynia would be reduced.

## 2. Materials and Methods

The study group consisted of 76 women aged 19 to 58 diagnosed with vulvodynia. The research was conducted from January to April 2022 using (1) the questionnaire technique. In order to obtain a detailed insight into the intensity of chronic pain in the vulva and its impact on the quality of life (QL), the authors of the study used several research tools to analyze individual aspects of life. An important aspect that was addressed in this group of patients was the biopsychosocial sphere, including sexual activity and satisfaction with health care. The authors wanted to examine areas that seem to be neglected and are particularly important in the context of problems in the daily functioning of these women. The used tools were (A) the author’s questionnaire (76 questions), (B) the WHOQOL-BREF questionnaire, and (2) the VAS.

The author’s questionnaire was constructed on the basis of the ProfiTest platform and made available together with the WHOQOL-BREF questionnaire and the VAS, by posting a direct link on three online support groups for women with vulvodynia. The inclusion criteria were female gender, diagnosis of vulvodynia by a doctor, and consent to be included in the study by completing a questionnaire. All women were informed of the aim of the research, anonymity, and confidentiality, and the possibility of withdrawing from participation at any stage. The study was conducted anonymously on a sample of volunteers in accordance with the guidelines of the Code of Ethics for Researchers and the Helsinki Declaration prepared by the World Medical Association [36,37].

### 2.1. The Author’s Questionnaire

The author’s questionnaire consisted of a total of 22 questions, both open and closed, and single and multiple choice. It was divided into thematic areas, which concerned an assessment of the sociodemographic, gynecological, and obstetrical data, health history and history of the disease, type of perceived pain in the vulvar area, experience of stress on a daily basis, and a subjective QL assessment. Patients made a subjective assessment of their stress level by marking one of the answers as “low”, “moderate”, or “high”. These categories were separated based on an analysis of the literature [38].

### 2.2. Visual Analogue Scale (VAS)

VAS is a simple tool that allows the respondent to subjectively determine the intensity of the perceived pain. It was used in this study to evaluate chronic vulval pain experienced while completing the survey. The respondents inserted a number (which did not have to be a whole number) based on a line drawing that marked the severity of pain on a scale of 0 to 10 (0 = no pain at all; 10 = the strongest pain imaginable) [39].

### 2.3. The World Health Organization Quality of Life (WHOQOL-BREF) Questionnaire

The WHOQoL-BREF questionnaire consists of 26 questions that assess respondents’ perception of quality of life, perception of their own health, and quality of life in four domains: physical health, psychological, social, and environmental.

In the first two, i.e., the perception of quality of life and health, QL is expressed on a scale of 1–5. The remaining domains are assessed on a scale of 4–20. Higher numbers denote a better quality of life. Perception of quality of life and perception of health consist of one question each from the questionnaire (questions 1 and 2, respectively). The results can be interpreted in accordance with the content of the answers to these questions. The other domains contain many questions, and there are no standards that determine which outcomes denote good or bad quality of life. However, since all domains are expressed on the same scale, it is possible to compare quality of life in particular domains. The authors chose the Polish adaptation of the short version of the WHOQoL-BREF questionnaire, which is considered a reliable, valid tool for assessing quality of life [40].

### 2.4. Statistical Analysis

An analysis of normality was performed using the Shapiro–Wilk test, and an analysis of quantitative variables was performed by calculating the mean, standard deviation, median, and quartiles. The analysis of qualitative variables was performed by calculating the numbers and percentages of occurrences of each value. The comparison of the values of the qualitative variables in the groups was performed using the chi-square test (with Yates’s correction for 2 × 2 tables) or the Fisher’s exact test, in which low expected frequencies appeared in the tables. The comparison of the values of quantitative variables in the two groups was performed using the Mann–Whitney test. The comparison of the values of quantitative variables in three or more groups was performed using the Kruskal–Wallis test. After detecting statistically significant differences, a post hoc analysis with Dunn’s test was performed to identify which groups statistically differed from others. Correlations between quantitative variables were analyzed using the Spearman correlation coefficient. The significance level of 0.05 was adopted in the analysis. The analysis was performed in the R software, version 4.1.3.

## 3. Results

A total of 76 women aged 19 to 58 were enrolled in the study (30.93 +/− 8.24). The study group was described in terms of individual and sociodemographic characteristics, as presented in Table 2. The majority of the women surveyed were between 26 and 35 years of age (46.06%), and had a weight between 51 and 60 kg (51.32%) and a height between 161 and 170 cm (60.53%). Considering the BMI index, the majority of female respondents (80.26%) were of a healthy weight, marginally fewer (10.53%) were overweight, and a small proportion (7.89%) were underweight. One woman (1.32%) was in a state of starvation. The majority of respondents came from large cities (46.05%), had higher education (69.74%), and declared unmarried status (52.63%).

Most respondents (65.79%) had never been pregnant. Among the rest, more than half (53.85%) had given birth once, and slightly less than half (38.46%) twice. The majority of women reported a vaginal delivery (68%). Cesarean section was reported by 28% of women. Only one respondent reported both modes of birth (Table 3).

The general assessment of the respondents’ health conditions, which were determined by comorbidities and the level of stress experienced on a daily basis, among other factors, is presented in Table 3. The majority of women struggled with bladder (39.47%) and intestinal (34.21%) problems, and temporomandibular joint popping and jaw clenching (39.47%). The majority of respondents assessed their level of daily stress as average (60.53%). Most of the surveyed women had received the vulvodynia diagnosis 0–5 years previous (72.37%), but in the majority of women (46.05%), the duration of vulval symptoms had been for more than 5 years, or between 1 and 5 years (44.74%) (Table 4).

When asked about symptoms in the area of the vulva, perineum, urethra, or anus that lasted for at least 3 months, the vast majority of respondents (72.37%) answered “Burning” followed by (56.58%) “Vulvar discomfort”. Only 3.95% of respondents did not report any complaints. Among the responses to “Other”, “Itching”, “Swelling”, “Arousal”, and “White discharge” were among the reported items (Figure 1).

The respondents mainly felt pain in the vulvar area during sexual intercourse (57.89%), when wearing underwear that was too tight (53.95%), and spontaneously, regardless of the situation (52.63%). The smallest group (3.95%) declared that all complaints had resolved spontaneously or as a result of therapy (physiotherapy, yoga, pelvic floor massage, psychotherapy, relaxation). “Other situations” in which pain appeared were a few days after intercourse or under the influence of stress and/or nerves (Figure 2).

The most common form of vulvodynia treatment implemented in respondents was physiotherapy (84.38%). Less common forms were pharmacological treatment and psychological assistance (both 45.31%) (Figure 3).

The majority of respondents (37.5%) considered that the treatment used until then had moderately reduced their symptoms. Marginally fewer women (23.44%) stated that the implementation of the treatment significantly reduced the symptoms, and only 4.69% declared that the implemented treatment completely reduced the symptoms (Figure 4).

When analyzing pain intensity on the VAS, where 0 denotes no pain and 10 the greatest imaginable pain, most women (23.68%) experienced symptoms at level 6 (Figure 5).

When asked “How did vulvodynia affect your quality of life on a scale from 0 to 5?”, where 0 denotes no impact on quality of life and 5 denotes a significant deterioration in quality of life, most women (64.47%) reported that vulvodynia significantly worsened their quality of life (Figure 6).

In women that stated that vulvodynia decreased their quality of life, the main causes were limitations related to their ability to perform activities of daily living and a decrease in satisfaction with sexual life (both 27.63%) (Figure 7).

Most often (44.74%), women assessed their quality of life as average (neither good nor bad). Least often, respondents assessed their quality of life as very good (6.58%) or very bad (5.26%) (Figure 8).

The quality of life of women with vulvodynia was described in accordance with the guidelines provided in the WHOQoL-BREF questionnaire. The domains of respondents’ quality of life are shown in Table 5. The respondents rated their quality of life best in the psychological domain, and marginally worse in the environmental and social domains. The physical domain had the lowest scores.

Analyses of the individual and sociodemographic data and the occurrence of individual aliments in women with vulvodynia were carried out. There were statistically significant differences between the occurrence of certain ailments in the vulvar area and the age, level of education, and marital status of the respondents. Among respondents, vulvar pain was more common in the age group up to 25 years (*p* = 0.002) (Table 6). No significant correlations were noted between BMI and the occurrence of individual complaints in the vulvar region (*p* > 0.05).

The relationship between the intensity of vulvar pain on the VAS and the level of stress experienced in everyday life was analyzed. The severity of pain was significantly higher (*p* = 0.045) in women who assessed their level of stress in their everyday life as medium or high as compared with women who assessed it as low. Those who rated their stress level as low experienced the lowest range of pain (4 +/− 1.26). Respondents that reported medium to high stress levels experienced the widest range of pain on the VAS (from 0 to 10) (Figure 9).

The severity of pain on the VAS correlated significantly and negatively with the respondents’ quality of life (*p* < 0.001; r = −0.447) (Figure 10a) and their health perception (*p* < 0.001; r = −0.483) (Figure 10b), indicating that the greater the pain experienced, the lower the perceived quality of life and health.

Among all the assessed quality-of-life domains, only the social domain was not significantly correlated with the experienced pain (*p* = 0.13; r = −0.173) (Figure 11c). The other three domains were significantly and negatively correlated with pain: physical (*p* = 0.03; r = −0.249) (Figure 11a), psychological (*p* = 0.023; r = −0.261) (Figure 11b), and environmental (*p* = 0.035; r = −0.243) (Figure 11d).

The authors also analyzed the correlations between the number of years passed since the vulvodynia diagnosis and quality of life, and health perception and each quality-of-life domain. Only the environmental domain results were significantly correlated with the time since diagnosis (*p* = 0.013; r = 0.287), showing that the quality of life in that domain improves with time (Figure 12).

In order to assess whether the use of any vulvodynia treatment had an impact on respondents’ quality of life, the Mann–Whitney test was used. Women who used any form of treatment evaluated their quality of life in the physical (*p* = 0.043) and psychological (*p* = 0.021) domains as being higher than those who did not report the use of any treatment (Figure 13).

Among the other treatments, including pharmacological and psychological treatments, only physiotherapy had an impact on one of the quality-of-life domains. Respondents who used physiotherapy as their treatment method assessed their quality of life in the psychological domain as being higher than those who did not use physiotherapy (*p* = 0.02) (Figure 14).

## 4. Discussion

Vulvodynia is a disease that affects all spheres of the life. Hence, taking a broader view of the problem and exploring solutions that could bring patients relief are vital. An analysis of the literature indicates that this study is pioneering and unique, as it is the first to address the quality of life of women with vulvodynia in Poland. The results of various studies clearly indicate that pain limits many areas of these patients’ lives. It often results in the abandonment of previous activities and social relationships, including those with loved ones. Disruption of one of the integral spheres of life can cause depression and isolation. Therefore, early diagnosis and new ways in which to comprehensively treat the disease and treat it broadly are essential. Therefore, the purpose of this study was to investigate the severity of chronic vulvar pain in women with vulvodynia and its impact on QL.

The sociodemographic characteristics reported in various studies [41,42,43,44] show that vulvodynia is prevalent in women under the age of 30. Other research indicates that more than 75% of female patients are less than 34 years old and most of them are under 25 years old [2,10,15,45,46]. In a study by Lua LL. et al. [47] on the latest treatments for vulvodynia in the U.S. involving 12,584 female respondents, the majority of sufferers were between 18 and 34. The results obtained are consistent with data in the literature. In fact, it was shown that most women with vulvodynia are under 25 years old, and the smallest group was composed of women over 35. Therefore, it can be concluded that the disease mainly occurs in young women, and that the differences between the various study results are due to the different groups in different populations.

By analyzing body weight and height, it was found that most respondents were women weighing between 51 and 60 kg and between 161 and 170 cm tall, i.e., with a normal BMI. No correlations were found in the literature regarding vulvodynia in the context of isolated parameters, but correlations are observed in the area of BMI. Indeed, Naess I and Bø K. [44] showed that vulvodynia appeared more often in individuals with a low BMI. Another study on the remission of the disease conducted by Nguyen RH. and Mathur C. et al. [48] indicated that women presenting as underweight or obese were significantly less likely to have it than those with a normal BMI. On the other hand, the results obtained by Naess I and Bø K. [49] show that the average BMI of people with chronic vulvar pain was normal and was approximately 22. The results of our study are consistent with these data. In light of the literature data, it can be concluded that vulvodynia occurs in women with varying BMIs, and elucidating the possible effects of being underweight or obese in the context of disease symptomatology will require further research.

When analyzing the educational level of female patients, an interesting correlation was observed. A study by Xie Y. et al. [27] indicated that, among female patients, the highest percentage was made up of those with a college education (88.41%). Similar results were obtained by Arnold LD. et al. [34] and Reed BD. et al. [9]. The former showed that the majority of women had either completed college or had a master’s degree. Our study results are in line with the global data, as the largest number of women had a college degree (69.74%). We suppose, therefore, that this result may be linked to the greater ambition and task-orientedness of more highly educated women, the greater pressure on themselves, the greater awareness of their health, and/or more frequent consultation with specialists who then diagnose vulvodynia. The high levels of induced stress correlated with the disease also seem to play an important role here.

Marital status does not appear to be related to the incidence of the disease. In a study by Ponte M. and Klemperer E. et al. [35], married women (51%) and single women (37%) accounted for the largest percentages. Similarly, in a study by Arnold LD. and Bachmann GA. et al. [34], most women reported a married status (64.9%). In contrast, in our study, the largest percentage of female respondents was composed of unmarried women (52.63%).

When performing an analysis on the gynecological–obstetric history of respondents, the authors of the present study noted that the largest number of women with vulvodynia had never been pregnant (65.79%), and of those who had given birth, most had done so once (53.85%), and this was most commonly a natural birth (68%). A study by Arnold LD. et al. [34], which also analyzed the medical history of patients, showed that the number of pregnancies was not correlated with the presence of vulvodynia, meaning that in the group of women without vulvar pain, women were more likely to report pregnancies than women with vulvodynia. A research paper by Nguyen RH. and Stewart EG. et al. [50], analyzing type of delivery, showed that women with vulvodynia were more likely to give birth by cesarean section, which may be due to pelvic floor dysfunction. Considering the data presented, it seems important to undertake further research focused on pregnancy and delivery in these women.

Vulvodynia often occurs with other comorbidities or dysfunctions. A study by Arnold LD. et al. [34] showed a threefold higher incidence of its co-occurrence with irritable bowel syndrome, chronic yeast infections, and urinary tract infections. Similar findings were presented by Xie Y. et al. [27], describing multimorbidity, such as irritable bowel syndrome, migraine headaches, temporomandibular joint dysfunction, cystitis, and endometriosis in these women. Our research confirmed the above data, as the largest group of women with vulvodynia reported bladder complaints and/or temporomandibular joint dysfunctions (39.47% each), followed by bowel problems and/or back pain (34.21% each). Single cases of fibromyalgia and endometriosis were also reported. It was also shown that vulvar discomfort was statistically significantly more frequent in women with intestinal disorders, with back pain and clenching, those who reported skipping jaws or grinding teeth, and in those with knee, ankle, or hip pain. The analyses presented here clearly indicate the importance of comorbidities in vulvodynia. The differences in the results presented may have been due to the dissimilarities of the study group and population variability.

There are a number of reports that clearly indicate that there is a correlation between high levels of stress in women and vulvodynia [6,7]. In addition, it should be noted that the disease is often observed in highly educated, sensitive, and task-oriented women, e.g., so-called businesswomen, with exposure to high amounts of stress [3,9,16,41,51], which is a significant predictor of vulvar pain [23,42,52]. Excessive sympathetic nervous system activity may lead to increased tension in the pelvic floor muscles, which is one of the possible causes of vulvar pain. However, Arnold LD. et al. [34], when investigating the correlations between stress levels and vulvodynia as compared to healthy subjects, found no significant differences in this regard. In the present study, female patients most often rated their stress level as medium (46%), highlighting the need for further research in this area.

In our study sample, the majority of women (72.37%) had received a vulvodynia diagnosis in the previous 5 years, but for 46.05% of participants, symptoms had been present for more than 5 years. This shows that a large percentage of women suffered from vulvar pain long before diagnosis. These results are in accordance with other studies. In the study by Trutnovsky et al. [53], history of vulvar pain ranged from 3 months to 20 years in the surveyed patients. Lamvu et al. [54] reported that the women in their study contacted a specialist around 2 years after the onset of vulvodynia symptoms.

To describe the discomfort in the vulvar area, female patients use many subjective terms. For example, Reed BD. and Harlow SD. et al. [9] showed that women most often describe pain as irritating (53.6%) and burning (51.4%). In a study by Arnold LD. et al. [34], “burning” was also the main descriptor of pain (88%). The results of our study also indicate that women are most likely to experience burning vulvar pain (72.37%). A study by Lamvu G. and Alappattu M. et al. [54] described the occurrence of moderate-to-severe pain associated with sexual intercourse. In addition, a clinical examination of the muscles of the vulvar area and the vaginal vestibule aggravated pain, but to a low degree. The results of our study are consistent with the cited data, as the majority of women reported the main trigger of vulvar pain to be sexual intercourse (57.89%). Thus, it can be concluded that dyspareunia is one of the most common problems in women with vulvodynia, and that it affects an extremely important sphere of human life. Therefore, there is a need for a comprehensive treatment and extensive therapy aimed at this aspect of the disease. The work of a multispecialist team that includes a psychologist and/or sexologist, among others, seems to be key for the holistic treatment of women with vulvodynia.

When considering methods and techniques for treating vulvodynia, the significant role of physiotherapy is clearly emphasized. Dionisi B. et al. [18] studied the effectiveness of these methods, with particular emphasis on methods such as biofeedback, TENS combined with electrical stimulation, and home therapy with pelvic floor stretching. They noted improvements in as many as 75.8% of the patients studied. On this basis, they concluded that pelvic floor relaxation with the combined use of physiotherapeutic methods is effective in the treatment of vulvodynia. A prospective cohort study by Brotto et al. [19] also showed the positive effect of multimodal physiotherapeutic therapy applied for 10–12 weeks on vulvar complaints, including dyspareunia and sexual functioning in women with provoked vulvodynia. Another study by Miletta M et al. [20] examined the effects of physiotherapy administered for 4 months in women diagnosed with vulvodynia. After this time, a 2-3-degree decrease in pain intensity on the VAS was determined. Zolnoun D. et al. [21] also reported similar conclusions. Moreover, our study has also confirmed that the frequent use of physiotherapy (84.38%) among women with vulvodynia clearly had a positive effect on reducing the negative sensation in the vulvar area and their QL. Thus, it can be concluded that physiotherapy is an effective method for alleviating these painful conditions, and should be widely used and developed, including in the context of educating medical personnel.

After analyzing the intensity of vulvar pain on a 10-point VAS in our study, it was shown that most women rated the intensity of vulvar pain as significant, indicating scores of 6 (23.68%), 5 (18.42%), and 7 (13.16%), with maximum pain ratings of 9 and 10 also being indicated (3.95% each). A study by Lamvu G. and Alappattu M. et al. [54] similarly found that women reported moderate to severe levels of general pain on the MPQ-Numeric pain scale. However, on clinical examination, the average pain in the vulvar area measured on the VAS was low (at the 2–3 score level). In another study by Pâquet M. and Vaillancourt-Morel. et al. [55], pain intensity was measured using the Numeric Rating Scale. They were asked to estimate the average vulvar and vaginal pain over the previous 6 months on a scale of 0 (no pain) to 10 (worst pain ever). The average was 7.15. Women with vulvodynia experience varying degrees of vulvar pain intensity in a variety of situations. It seems reasonable, therefore, to study its intensity in specific conditions, identify them, and collate these studies.

Our study analyzed the influence of sociodemographic characteristics on the nature of presented complaints. It was shown that vulvar pain occurred significantly more often in the group of women under 25 years of age, i.e., the youngest group. The same symptoms were also most common in respondents with a high school education and in those who were unmarried. No significant differences were found when comparing female respondents in terms of their body weight and height.

Our research indicated that there was a relationship between stress and vulvar pain and its severity. Pain on the VAS was significantly higher in those women rating the level of stress in daily life as medium (60.53%) or high (31.58%) than in those rating it as low (7.89%). This relationship seems to be confirmed by a study by Chisari C. et al. [5] who found that stress may be a significant predictor of pain in women with vulvodynia. This study was cross-sectional, and its purpose was to test the effect of a number of factors, including stress, on the severity of vulvar pain. It showed in a statistically significant manner that lower belief in treatment control, catastrophic thinking, stress, and greater identification with the disease were significant factors that could cause pain.

QL is a multidimensional term covering a number of aspects of human life, including emotional state and mental condition, physical health, and social functioning. The presence of an illness, especially a chronic one, can significantly limit a person’s functioning, significantly reducing his or her QL.

The quality of life of patients with vulvodynia is a topic that is not often discussed in the literature. In a study by Arnold LD. et al. [34], they used a modified Ladder of Life scale, and the data obtained indicated that both women with vulvodynia and healthy women reported similar levels of stress, but vulvodynia patients exhibited a worse overall QL. Moreover, up to 42% of them felt their lives were out of control. In contrast, in the study by Lamvu et al. [54], vulvodynia patients rated their quality of life as being best in the physical function domain and in the pain domain. In another study by Ponte M. et al. [35], which included 280 patients from a university vulvodynia clinic, 101 of whom were diagnosed with vulvodynia and the rest with other vulvar diseases, a condition-specific Skindex-29 scale was used to assess QL. The study showed that women with vulvodynia had a significantly lower QL than those with vulvar dermatological diseases. When analyzing the individual domains, the physical and emotional aspects in both groups of subjects were rated similarly, in contrast to the functioning domain, in which a worse QL was reported by women with the disease. In addition, these patients reported vulvar pain that occurred frequently or always, resulting in depression or frustration. Furthermore, the study showed its significant impact on social functioning, concerning, for example, difficulties in pursuing passions or doing work. It also noted a greater sense of loneliness and a concern that the vulvar condition could affect the ability to be part of a happy relationship. A study by Xie Y. et al. [27] on the QL and economic burden of women with vulvodynia in the U.S., using the Euro QOL 5 dimensions scale (EQ-5D), is in agreement with the previously cited data. They reported a low QL with an emphasis on social isolation and inability to participate in social life. The results of our study are consistent with those of the aforementioned study. Women with vulvodynia rate their QL as average (neither good nor bad), with it being best in the psychological domain and worst in the physical domain. Moreover, the majority of respondents (64.47%) declared that vulvodynia significantly decreased their quality of life, rating its influence as 5 on a scale from 1 to 5. However, nearly half of the surveyed women assessed their general quality of life as average, suggesting that, despite the major negative impact of vulvodynia, their quality of life remained average. The women surveyed scored the highest scores in the psychological and environmental domains of the WHOQoL-BREF questionnaire. Although the women surveyed scored the highest in the psychological domain, some studies show that women with vulvodynia suffer from anxiety and depression [56,57]. In a Spanish study by Tribó et al. [56], 61.7% of vulvodynia patients reported suffering from anxiety and 24.7% were depressed. Their mean quality-of-life score as assessed by the DLQI (Dermatology Life Quality Index) was 12.9, with the maximum score being 30. In the statistical analysis, anxiety was negatively correlated with QL for the women surveyed, indicating that the higher the anxiety, the lower their perceived quality of life. Arnold et al. [57] showed that chronic vulvar pain in a sample of women with vulvodynia was associated with significantly greater odds of depression (OR = 2.99). Women with vulvodynia reported higher stress levels and lower quality of life than those from the control group; however, the differences were statically insignificant.

In our study, the authors analyzed the impact of factors such as pain, time since diagnosis, pain-aggravating factors, and treatment on the perceived quality of life of patients. We found that pain was significantly negatively correlated with patients’ quality of life in all domains, except the social domain. Women who experienced greater pain had a poorer quality of life. On the other hand, years passed since the vulvodynia diagnosis correlated positively with quality of life, but the correlation was significant only in the environmental domain. In the study by Naetss et al. [58], the quality of life of patients with provoked vestibulodynia was compared to a healthy control group using the SF-36 questionnaire. Healthy controls had statistically significantly higher scores in every QOL domain, except the physical functioning and emotional domains, indicating that chronic vulvar pain decreases quality of life.

The authors attempted to analyze the impact of pain-intensifying situations on quality of life and found that there were no significant differences between perceived quality of life in all domains in women who experience such situations and those who do not. This might indicate that, despite the different situations in which respondents felt pain in the vulvar area, their quality of life was similar.

Lamvu et al. [51] found that women with vulvodynia who received at least one type of treatment reported an improvement in quality-of-life domains, including physical limitations, emotional limitations, well-being, and social function after 6 months. Those results are in accordance with our findings. Respondents who received any type of vulvodynia treatment had significantly higher scores in the physical and psychological domains. Furthermore, those who received physiotherapy treatment had significantly higher scores in the psychological domain than those who did not receive physiotherapy treatment. This may indicate that physiotherapy has a positive influence on the mental health of vulvodynia patients. According to various studies, the methods used by physiotherapists alleviate discomfort, and thus have a positive effect on the QL of female patients. Forth HL. et al. [22] assessed the impact of a 3-month period of physiotherapy treatment on the QL, among other things, of women with vulvodynia using the abbreviated SF-36 questionnaire (version 2), covering eight aspects of health: physical functioning, physical aspect, perceived pain, general health, vitality, social functioning, and the role of emotional and mental health. After physiotherapy, QL was found to improve in areas such as bodily pain, mental health, social functioning, and emotions. The psychological sphere improved to a greater degree than the physical area. The results were not statistically significant, most likely due to the small study group, but they did indicate a trend. Our study showed that many women with vulvodynia benefit from a physiotherapist (84.38%). Moreover, its positive effect on QL was noted, especially in the psychological sphere, where it was significantly better as compared to the other forms of treatment.

Our study on the impact of chronic vulvar pain on the QL of women with vulvodynia also has some limitations. As a result of reduced access to women with diagnosed vulvodynia, the study group consisted of 76 subjects. The limited number of analyses means that there is an increased chance of a Type I error. We intend to continue our work and undertake a broader analysis that includes the Polish population, additionally including detailed studies on sexual activity and universally available multispecialty care and its limitations. Such a study may explain why the QL of these patients is average (neither good nor bad), and the use of other standardized research tools would make it possible to expand the study to include certain spheres that we consider neglected. However, given these limitations, it should also be emphasized that, to the best of our knowledge, this is the first study to assess the quality of life in the context of chronic pain of women with vulvodynia in the Polish population.

An additional limitation of the study is the lack of diversity in the group of women. As we have noted, our study mainly included women from large cities and with a higher educational status. There is an assumption that if patients from smaller towns and with a lower educational status were included, the quality of life reported would be at a lower level, because, in this group, the availability of appropriate treatments for vulvodynia and general awareness of the disease would be lower.

The choice of the WHOQOL-BREF questionnaire could be seen as a weakness of the study, as we did not find publications in which women with vulvodynia were examined for HRQoL using this tool. If we had chosen to use another scale that other authors had used before (e.g., the Ladder of Life scale, Skindex-29, Euro QOL 5 dimensions scale (EQ-5D)), our quality-of-life comparisons would have been more accurate. On the other hand, we think that the choice of the WHOQoL-BREF questionnaire may have been an advantage. This new tool very accurately examines individual spheres of quality of life, considering additional aspects such as relationships with people, sexual life, and satisfaction with health care institutions, which, as we know, strongly affect the quality of life of women with vulvodynia.

## 5. Conclusions

In this study, we found that pain was mainly reported by younger women. This may be because younger women are more likely to receive a diagnosis of vulvodynia, which is associated with greater body awareness, increased attention to health, and more frequent consultation with specialists such as gynecologists. Various other studies in other populations also noted an association between age and the incidence of complaints concerning the vulvar region, which is important for further studies. However, no association was found concerning personal characteristics such as body weight and height, or sociodemographic characteristics such as place of residence. Our study also shows that stress is significantly associated with vulvodynia complaints. The severity of pain was shown to be higher in women whose stress levels were described as medium or high. The association of stress with excessive pelvic floor tension has already been demonstrated in other studies, confirming our hypothesis. The pain reported by women with vulvodynia was also shown to have a negative impact on quality of life, which, in our study, was most often described as average (neither good nor bad). Negative impacts were mostly associated with restrictions related to being able to perform activities of daily living and engage in sexual activity, which are included in the basic components of a person’s quality of life. To the best of our knowledge, our study of the female population in Poland is the first of its kind, and future studies should focus on a larger number of patients. In conclusion, our results will help health care professionals to take a broader, more holistic view of women with vulvodynia and seek solutions that do not limit the spheres discussed. This could include case-by-case analyses of female patients with vulvodynia within a wider diagnostic and therapeutic team, focusing on problems that interfere with patients’ daily lives, and further educating health care professionals on the subject.

## Figures and Tables

**Figure 1 life-13-00328-f001:**
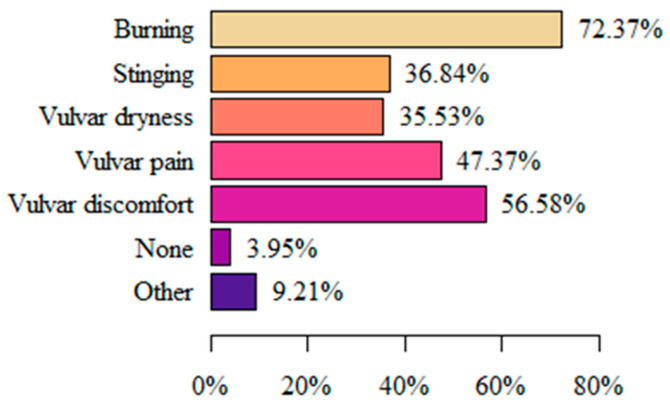
Symptoms related to vulvodynia.

**Figure 2 life-13-00328-f002:**
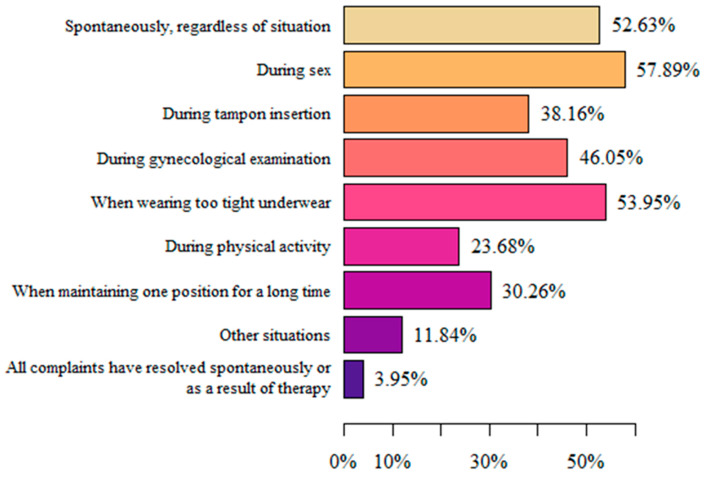
Situations in which pain complaints related to vulvodynia appeared.

**Figure 3 life-13-00328-f003:**
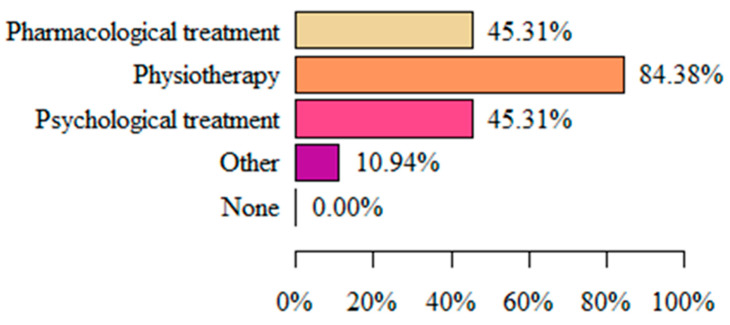
Type pf vulvodynia treatment.

**Figure 4 life-13-00328-f004:**
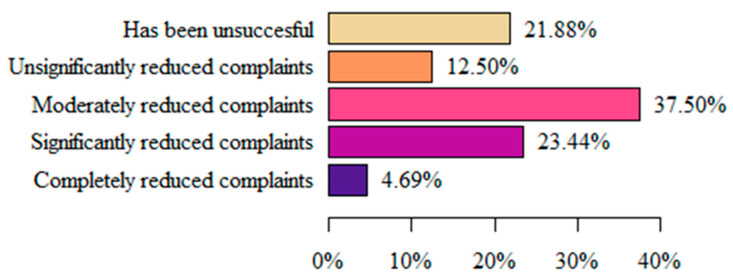
Respondents’ assessment of the effectiveness of the applied treatment.

**Figure 5 life-13-00328-f005:**
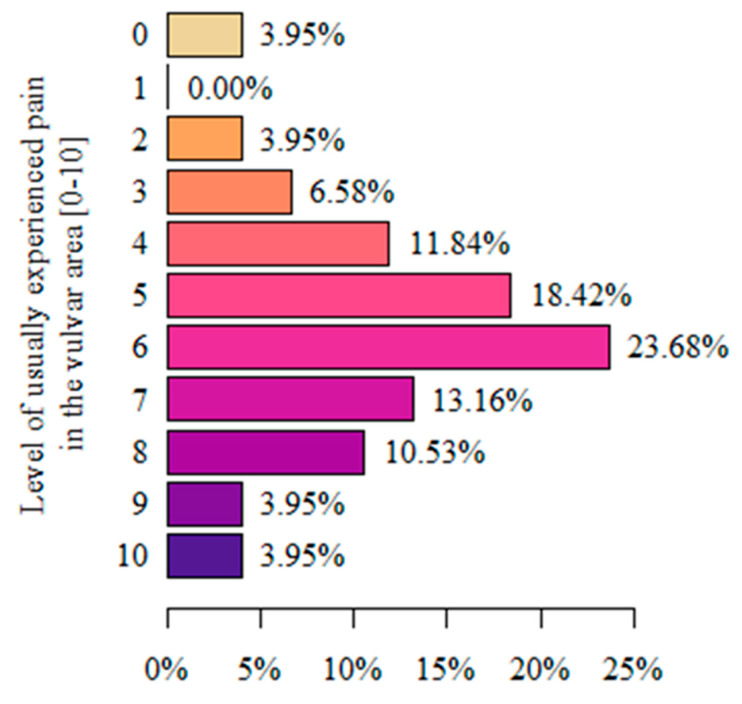
Assessment of the pain intensity in the vulvar area using the VAS.

**Figure 6 life-13-00328-f006:**
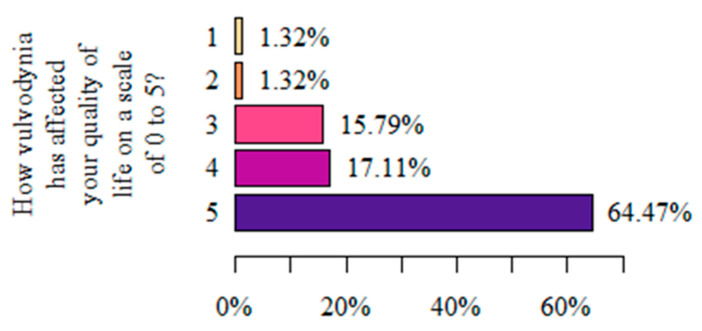
Respondents’ assessment of the impact of vulvodynia on the perceived quality of life (authors’ questionnaire).

**Figure 7 life-13-00328-f007:**
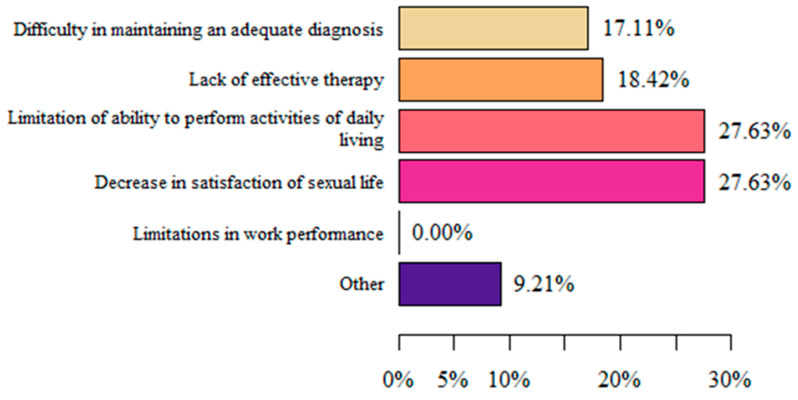
Respondents’ reasons for decreased quality of life (authors’ questionnaire).

**Figure 8 life-13-00328-f008:**
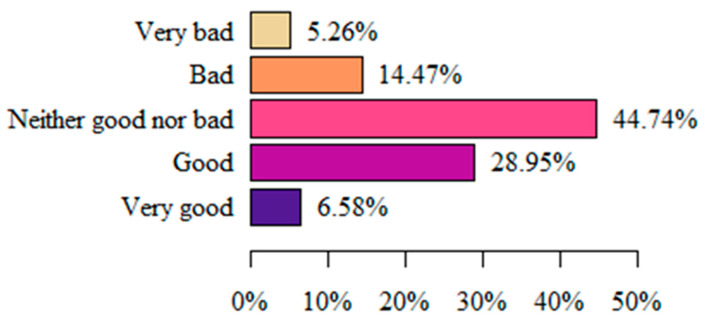
Respondents’ perception of quality of life (authors’ questionnaire).

**Figure 9 life-13-00328-f009:**
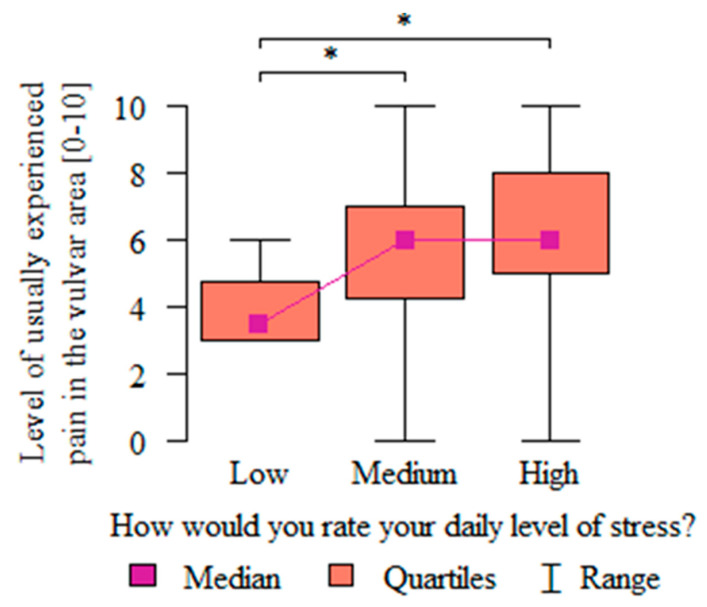
The relationship between the respondents’ daily stress level and pain intensity. * Statistically significant differences (*p* < 0.05) measured using Dunn’s post hoc test.

**Figure 10 life-13-00328-f010:**
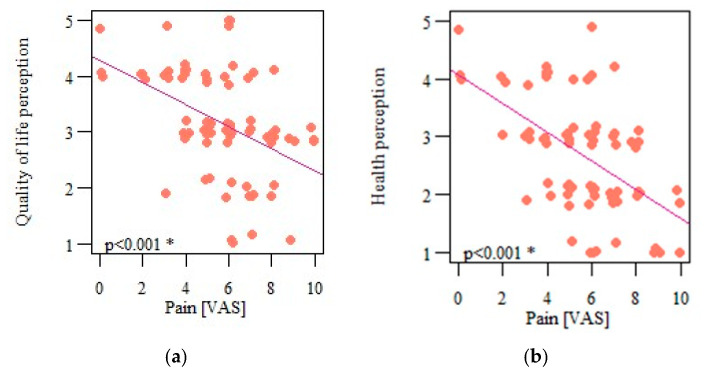
(**a**) Correlation of respondents’ perception of quality of life and pain intensity; (**b**) correlation of respondents’ health perception and pain intensity. * Statistically significant differences (*p* < 0.05).

**Figure 11 life-13-00328-f011:**
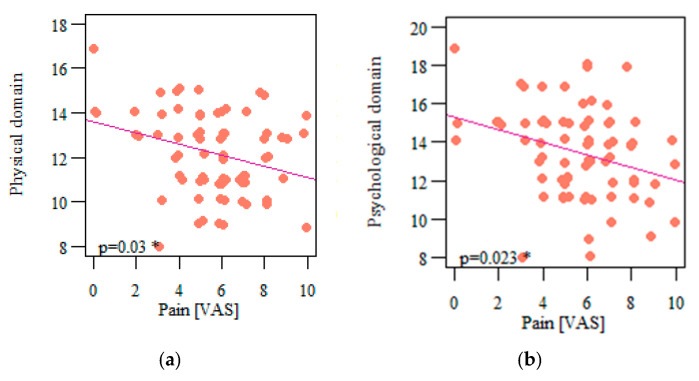
(**a**) Correlation of respondents’ quality of life in the physical domain and pain intensity; (**b**) correlation of respondents’ quality of life in the psychological domain and pain intensity; (**c**) correlation of respondents’ quality of life in the social domain and pain intensity; correlation of respondents’ quality of life in the environmental domain and pain intensity (**d**) * Statistically significant differences (*p* < 0.05).

**Figure 12 life-13-00328-f012:**
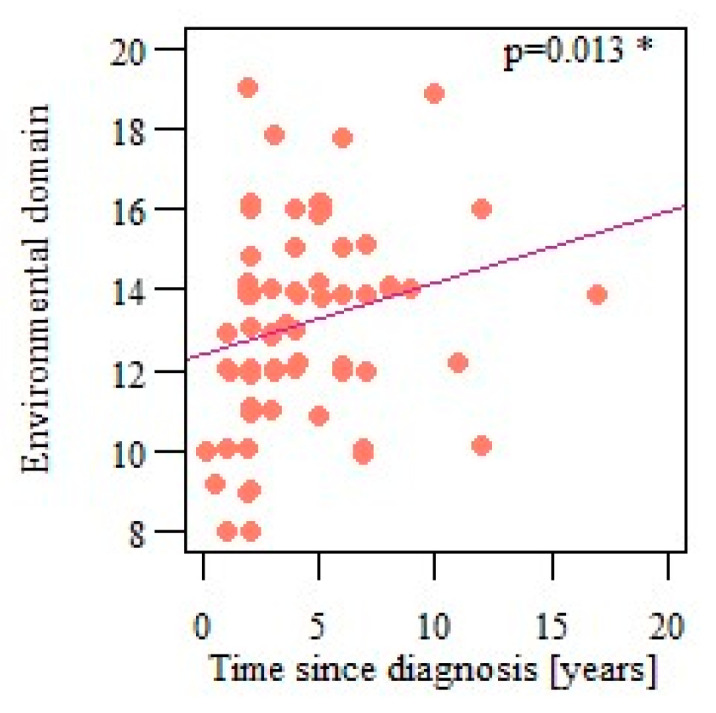
Correlation of respondents’ quality of life in the environmental domain and the time since vulvodynia diagnosis. * Statistically significant differences (*p* < 0.05).

**Figure 13 life-13-00328-f013:**
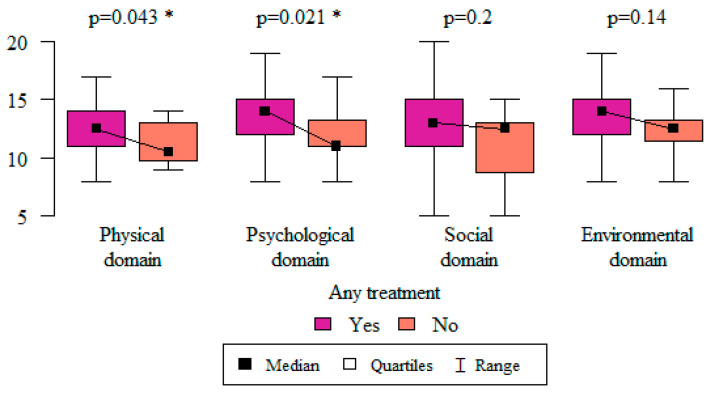
The relationship between the use of any form of treatment and the quality-of-life domains. * Statistically significant differences (*p* < 0.05) measured using the Mann–Whitney test.

**Figure 14 life-13-00328-f014:**
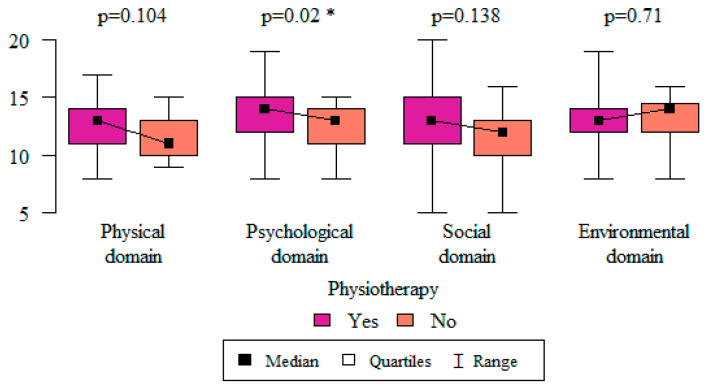
The relationship between the use of physiotherapy treatment and quality-of-life domains. * Statistically significant differences (*p* < 0.05) measured using the Mann–Whitney test.

**Table 1 life-13-00328-t001:** Organic causes of chronic vulvar pain (own elaboration based on [2]).

Factors	Characteristics
Infectious	Candida Herpes
Inflammatory	Lichen sclerosis Lichen planus Immune disorders
Neoplastic	Paget disease Squamous cell carcinoma
Neurological	Postherpetic neuralgia Pressure or injury to peripheral nerves Neuroblastoma
Traumatic	Episiotomy Obstetric procedures
Traumatic	Post-surgery, chemotherapy, radiation therapy
Hormonal	Genitourinary syndrome during menopause (vulvovaginal atrophy) Lack of menstruation during lactation

**Table 2 life-13-00328-t002:** Sociodemographic data of the respondents.

Variable	Subgroups	N (%)
Age	15–25	22 (28.95)
	26–35	35 (46.05)
	36–45	14 (18.42)
	46–55	4 (5.26)
	56–60	1 (1.32)
Weight in kg	40–50	12 (15.79)
	51–60	39 (51.32)
	61–70	22 (28.95)
	71–80	2 (2.63)
	81–90	1 (1.32)
Height in cm	140–150	1 (1.32)
	151–160	13 (17.11)
	161–170	46 (60.53)
	171–180	15 (19.74)
	181–190	1 (1.32)
BMI (kg/m^2^)	<16.0 (starvation)	1 [1.32]
	16.0–16.9 (emaciation)	0 [0]
	17.0–18.5 (underweight)	6 [7.89]
	18.5–24.9 (healthy weight)	61 [80.26]
	25.0–29.9 (overweight)	8 [10.53]
	≥30.0 (obesity)	0 [0]
Place of residence	Village	11 (14.47)
	City under 20,000 citizens	7 (9.21)
	City 20,000–100,000 citizens	23 (30.26)
	City over 100,000 citizens	35 (46.05)
Educational level	Elementary	1 (1.32)
	High school	11 (14.47)
	Vocational education	11 (14.47)
	Higher education	53 (69.74)
Marital status	Single	40 (52.63)
	Married	33 (43.42)
	Divorced	2 (2.63)
	Widowed	1 (1.32)

**Table 3 life-13-00328-t003:** Gynecological and obstetrical data.

Variable		N (%)
Pregnancies	Never been pregnant	50 (65.79)
	1	11 (14.47)
	2	9 (11.84)
	3	4 (5.26)
	4	2 (2.63)
Births	Never gave birth	1 (3.85)
	1	14 (53.85)
	2	10 (38.46)
	3	1 (3.85)
Mode of births	Vaginal	17 (68)
	Cesarean section	7 (28)
	Both	1 (4)

**Table 4 life-13-00328-t004:** Assessment of the health of the group.

Variable		N (%)
Comorbidities	Bladder problems	30 (39.47)
	Intestinal problems	26 (34.21)
	Back pain	26 (34.21)
	Jaw clenching, temporomandibular joint popping, teeth grinding	30 (39.47)
	Headaches	17 (22.37)
	Hip, knee, or ankle pain	13 (17.11)
	Other	8 (10.53)
	None	11 (14.47)
Perceived stress level	Low	6 (7.89)
	Moderate	46 (60.53)
	High	24 (31.58)
Years passed since the diagnosis of vulvodynia	0–5	55 (72.37)
	10–6	15 (19.74)
	15–11	3 (3.95)
	16–20	1 (1.32)
	No given answer	2 (2.63)
Duration of discomfort in the vulvar area	Less than 6 months	3 (3.95)
	6 months–1 year	4 (5.26)
	1–5 years	34 (44.74)
	Longer than 5 years	35 (46.05)

**Table 5 life-13-00328-t005:** Characteristics of quality of life on the basis of individual domains of the WHOQOL-BREF questionnaire.

WHOQOL-BREF	Mean	SD	Median	Min	Max	Q1	Q3
Physical domain	12.18	1.87	12	8	17	11	14
Psychological domain	13.46	2.33	14	8	19	12	15
Social domain	12.72	3.35	13	5	20	11	15
Environmental domain	13.18	2.39	13	8	19	12	14

**Table 6 life-13-00328-t006:** Assessment of the quality of life in particular domains of the WHOQOL-BREF questionnaire in cases of increasing vulvar pain.

			WHOQOL-BREF					
Situations in Which Vulvar Pain Intensifies		N	QOL Perception	Health Assessment	Physical	Psychological	Social	Environmental
Spontaneously	Yes	40	3 ± 0.93	2.48 ± 0.91	12.22 ± 1.6	13.25 ± 2.3	13.15 ± 2.9	12.8 ± 2.15
	No	36	3.36 ± 0.93	2.86 ± 1.05	12.14 ± 2.09	13.69 ± 2.39	12.25 ± 3.75	13.61 ± 2.6
	*p*		0.128	0.123	0.665	0.392	0.211	0.180
During sex	Yes	44	3.8 ± 0.99	2.52 ± 1.02	12.09 ± 1.76	13.5 ± 2.42	12.48 ± 3.17	13.18 ± 2.47
	No	32	3.16 ± 0.88	2.84 ± 0.92	12.31 ± 2.02	13.41 ± 2.26	13.06 ± 3.59	13.19 ± 2.32
	*p*		0.762	0.202	0.657	0.852	0.644	0.936
During tampon insertion	Yes	29	3.17 ± 1.04	2.41 ± 1.02	12 ± 1.75	13.66 ± 2.26	12.9 ± 3.22	13.21 ± 2.3
	No	47	3.17 ± 0.89	2.81 ± 0.95	12.3 ± 1.94	13.34 ± 2.4	12.62 ± 3.45	13.17 ± 2.47
	*p*		0.973	0.067	0.525	0.693	0.689	0.849
During gynecological examination	Yes	35	3.09 ± 0.89	2.46 ± 0.95	12.06 ± 1.75	13.57 ± 2.17	12.06 ± 3.05	12.86 ± 1.97
	No	41	3.24 ± 0.99	2.83 ± 1	12.29 ± 1.98	13.37 ± 2.49	13.29 ± 3.52	13.46 ± 2.69
	*p*		0.372	0.084	0.611	0.996	0.084	0.322
When wearing underwear that is too tight	Yes	41	3.07 ± 1.06	2.61 ± 1.05	12.02 ± 1.74	13.41 ± 2.28	12.88 ± 3.47	13.07 ± 2.48
	No	35	3.29 ± 0.79	2.71 ± 0.93	12.37 ± 2.02	13.51 ± 2.43	12.54 ± 3.23	13.31 ± 2.31
	*p*		0.451	0.748	0.462	0.748	0.573	0.723
During physical activity	Yes	18	2.94 ± 0.87	2.28 ± 0.96	12.11 ± 2	13.22 ± 1.66	12.61 ± 2.64	12.44 ± 2.06
	No	58	3.24 ± 0.96	2.78 ± 0.97	12.21 ± 1.84	13.53 ± 2.51	12.76 ± 3.56	13.41 ± 2.46
	*p*		0.255	0.068	1	0.604	1	0.135
When maintaining one position for a long time	Yes	23	2.91 ± 1	2.43 ± 0.99	12.17 ± 1.77	13.39 ± 2.02	12.74 ± 2.93	13.17 ± 1.99
	No	53	3.28 ± 0.91	2.75 ± 0.98	12.19 ± 1.92	13.49 ± 2.48	12.72 ± 3.54	13.19 ± 2.56
	*p*		0.112	0.205	0.977	0.681	0.685	0.927
Other situations	Yes	9	3.44 ± 0.88	2.67 ± 0.87	12.44 ± 1.51	14.33 ± 1.32	12 ± 3.87	13.22 ± 2.77
	No	67	3.13 ± 0.95	2.66 ± 1.01	12.15 ± 1.92	13.34 ± 2.42	12.82 ± 3.29	13.18 ± 2.36
	*p*		0.407	0.946	0.677	0.121	0.421	0.922

## Data Availability

Data available on request due to restrictions privacy.

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
