# Peer review of "Chronic Vulvar Pain and Health-Related Quality of Life in Women with Vulvodynia"

_life, 2023, doi:10.3390/life13020328_

Round 1
Reviewer 1 Report
This study supports novel information about Chronic vulvar pain and quality of life in women with vulvodynia
This is an interesting topic and should be of great interest to the journal readers’. However, from my point of view, authors should include the following requeriments
After the review and in my humble opinion, the manuscript presents major problems. Below I present my recommendations separated by sections. Hopefully they will be useful:
The English needs to be reviewed by an English speaker. Some parts were not easy to understand. Several grammar mistakes.
First of all, I suggest change the section title, reflecting methods better, due to the fact, I believe it can be considerer an assessment a HRQoL as a suggestion
On the other hand, Introduction section may be improved adding new information in order to reflect better the state-of-art,regarding HRQoL For instance, I suggest to refers to VAS on foot disabilities as a physiotherapy field.
DOI10.3390/ijerph15102205
Methods are nwell-designed with relevant and complete information. Due to the fact that there are a good description of the properties of the outcome review as well as detailed statistical analyses were included.However, even though,Tables and, figures does reflects the main finding of the study. Authors should increase their statistical analysis to increase the evidence level of their achievement, I suggest to include R Spearman correlation instead of Fisher’s test. In addition, the results does not appears,in a proper way.
Disussion: Finally, as regards to discussion section, authors should compare their achievement with other similar prior researches in HRQoL. For that reason I suggest to include the following reference:
DOI:10.7150/ijms.48705
Furthermore, author should relate their results as regards to other VAS researches on the foot and reliabilty techniques. I suggest to include the following reference to complete this comment:
DOI:10.3390/ijerph15102205
Author Response
1 reviewer's comment: The English needs to be reviewed by an English speaker. Some parts were not easy to understand. Several grammar mistakes.
We sincerely thank you for the notice. The text has been checked and corrected by a translator from MDPI.
2 reviewer's comment: First of all, I suggest change the section title, reflecting methods better, due to the fact, I believe it can be considerer an assessment a HRQoL as a suggestion.
We have changed the section title from “Chronic vulvar pain and quality of life in women with vulvodynia” to “Chronic vulvar pain and health-related quality of life in women with vulvodynia” (line 2).
3 reviewer's comment: On the other hand, Introduction section may be improved adding new information in order to reflect better the state-of-art, regarding HRQoL. For instance, I suggest to refers to VAS on foot disabilities as a physiotherapy field. DOI10.3390/ijerph15102205
Thank you for your notice. We have included above survey to the Introduction section with several other studies that describe the use of the VAS scale to assess HRQoL in physiotherapy (lines 152 to 156).
4 reviewer's comment: Methods are nwell-designed with relevant and complete information. Due to the fact that there are a good description of the properties of the outcome review as well as detailed statistical analyses were included.However, even though,Tables and, figures does reflects the main finding of the study. Authors should increase their statistical analysis to increase the evidence level of their achievement, I suggest to include R Spearman correlation instead of Fisher’s test. In addition, the results does not appears,in a proper way.
As recommended by the Reviewer, we consulted a professional statistician, who gave his opinion that Fisher's test and Spearman's R test are used to analyze two different types of variables. Fisher's test is used to analyze the relationship between qualitative variables (qualitative, that is, those that are not numbers), and Spearman's R is used to analyze the relationship between quantitative variables (quantitative, that is, those that are numbers). So you can't swap one for the other.
5 reviewer's comment: Disussion: Finally, as regards to discussion section, authors should compare their achievement with other similar prior researches in HRQoL. For that reason I suggest to include the following reference: DOI:10.7150/ijms.48705
We sincerely thank you for your notice, but in this case we think that the topic of hemophilia and chronic vulvar pain are so far apart that we do not see the relevance of including this article in the paper.
6 reviewer's comment: Furthermore, author should relate their results as regards to other VAS researches on the foot and reliabilty techniques. I suggest to include the following reference to complete this comment: DOI:10.3390/ijerph15102205
As recommended earlier, in the introduction we referred to this article and others that pointed out the important role of the VAS scale for assessing HRQoL (line from 152 to 156).
Reviewer 2 Report
Below are my comments
Introduction
While you did a great job with providing a literature review, you need to end your introduction by providing the objective of the study and the hypotheses.
Methods
Did you conduct an a priori power analysis?
How was the VAS scale administered? You stated that they could have numbers not as integers? Did you use a line? A slider? Did you administer it on paper? Online?
How did you measure stress?
Also by author's own survey, it seems that you used a validay survey for QOL, so you should remove that from the author's own survey.
Based on what you have written in your statistical analysis section it seems that you did check for normality. Please make sure you clarify that you checked for normality and also what techniques, if any, you used to try to
Results
While the presentation of the data is excellent, I have several concerns about the results.
Based on the figures that you have presented with your results (especially Figure 12), I believe that some of your results may be a result of skewed data.
Another point of concern is the sheer number of analyses completed as this could result in a higher likelihood of a Type I error.
Discussion
I love paragraph 2, and I think it can be combined with paragraph 1 to make a very strong beginning of the discussion section
What do you mean that vulvodynia do occur in BMI?
Since you talk about BMI in your discussion, can you please provide the mean+/-SD for BMI and the distribution of the BMI data in your results section
I think on likes 593-603, you are trying to address one of the limitations of the study. Please include the other limitations of the study as well.
Author Response
1 reviewer’s comment: Introduction
While you did a great job with providing a literature review, you need to end your introduction by providing the objective of the study and the hypotheses.
We sincerely thank you for your comment. The introduction has been corrected - finished by stating the objective of the study and the hypothesis (line from 160 to 163).
2 reviewer’s comment: Methods
Did you conduct an a priori power analysis?
We sincerely thank you for your comment. At the planning stage of the study, a priori power was not analyzed. Convenience sampling was used, which means that the time and spatial frame of the study was established beforehand, and the responses of all patients available at that time were collected for analysis.
How was the VAS scale administered? You stated that they could have numbers not as integers? Did you use a line? A slider? Did you administer it on paper? Online?
The VAS scale was graphically represented as a line. The beginning of the line (point 0) and the end of the line (point 10) were described respectively as "no pain" and "greatest imaginable pain". Respondents were asked to rate the intensity of the pain that they experienced on the VAS scale by inserting any (including an incomplete) number in the indicated box. I have provided the scale online.
How did you measure stress?
Stress was measured through an objective assessment of female respondents, who were asked to specify the level of daily stress among 3 levels of "low," "moderate," "high." It was a part of the author’s questionnaire..
Also by author's own survey, it seems that you used a validay survey for QOL, so you should remove that from the author's own survey.
We sincerely thank you for this notice. It should be clarified that we assessed quality of life using 2 separate tools. We measured QL using the validated WHOQoL-BREF questionnaire, but we also used additional questions included in our own survey (subjective quality of life assessment) as a supplement.
Based on what you have written in your statistical analysis section it seems that you did check for normality. Please make sure you clarify that you checked for normality and also what techniques, if any, you used to try to
Yes, we have checked for normality by Shapiro-Wilk test.
3 reviewer’s comment: Results
While the presentation of the data is excellent, I have several concerns about the results.
Based on the figures that you have presented with your results (especially Figure 12), I believe that some of your results may be a result of skewed data.
Another point of concern is the sheer number of analyses completed as this could result in a higher likelihood of a Type I error.
We sincerely thank you for the right comment. We have once again carefully reviewed the results and removed Tables 6, 7 and 8 due to the small size of the study group.
4 reviewer’s comment: Discussion
I love paragraph 2, and I think it can be combined with paragraph 1 to make a very strong beginning of the discussion section.
Thank you very much for the positive perception of the initial paragraphs of the discussion. According to the comment, paragraph 1 and 2 have been combined (lines from 381 to 391).
5 reviewer’s comment:
What do you mean that vulvodynia do occur in BMI?
Since you talk about BMI in your discussion, can you please provide the mean+/-SD for BMI and the distribution of the BMI data in your results section.
We sincerely thank you for your comment. We were referring to the occurrence of analysis between BMI and vulvodynia. According to the right comment, we have included the distribution of BMI in the study group (Table 2). We also analyzed the relationship of BMI with the incidence of vulvodynia among the female respondents, but there were no statistically significant correlations.
6 reviewer’s comment: I think on likes 593-603, you are trying to address one of the limitations of the study. Please include the other limitations of the study as well.
Thank your for your notice. We have added additional limitations of the study (lines 530 to 545).
Round 2
Reviewer 1 Report
authors have adressed all my requeriments in the correct way
Author Response
Authors have adressed all my requeriments in the correct way.
We sincerely thank you for many valuable comments.
Reviewer 2 Report
I appreciate the authors addressing my concerns. I just have some minor concerns that should be addressed in the limitations section
1. Based on the distribution of the data in Figure 11, it seems that most of your data is concentrated near the bottom left and that is what might be causing a significant finding instead of a true correlation (explains the low R value)
2. Considering the sheer number of analyses that you have completed, there is an increased chance of a Type I error. Please make sure you address that in your limitations section.
Author Response
I appreciate the authors addressing my concerns. I just have some minor concerns that should be addressed in the limitations section
1. Based on the distribution of the data in Figure 11, it seems that most of your data is concentrated near the bottom left and that is what might be causing a significant finding instead of a true correlation (explains the low R value).
We sincerely thank you for your insightful analysis. The situation presented above required us to use non-parametric methods.
2. Considering the sheer number of analyses that you have completed, there is an increased chance of a Type I error. Please make sure you address that in your limitations section.
We sincerely thank you for this comment. We have clearly indicated this aspect in the limitations of the study (line 621-622).
